# The Phylogenetic Relationships of the Family Sciaenidae Based on Genome-Wide Data Analysis

**DOI:** 10.3390/ani12233386

**Published:** 2022-12-01

**Authors:** Xiaolu Han, Shihuai Jin, Zhiqiang Han, Tianxiang Gao

**Affiliations:** Fishery College, Zhejiang Ocean University, Zhoushan 316002, China

**Keywords:** phylogenomics, orthologous, Sciaenidae, genome-wide, phylogeny

## Abstract

**Simple Summary:**

Phylogenomics is an effective method to resolve phylogenetic problems. Herein, the phylogenetic relationships among 12 Sciaenidae species were reconstructed using genome-wide data. Two major phylogenetic groups were distinguished in our study, namely, the Western Atlantic species group, which contained four species, and the Eastern Atlantic, Indian Ocean, and West Pacific species group, which contained eight species. The Western Atlantic species were indicted as the more ancient group in the studied species. Meanwhile, the Eastern Atlantic, Indian Ocean, and West Pacific species were the younger group. However, a close phylogenetic relationship between the *Collichthys* and *Larmichthys* genera was not supported.

**Abstract:**

Uncertainty and controversy exist in the phylogenetic status of the Sciaenidae family because of the limited genetic data availability. In this study, a data set of 69,098 bp, covering 309 shared orthologous genes, was extracted from 18 genomes and 5 transcriptomes of 12 species belonging to the Sciaenidae family and used for phylogenetic analysis. The maximum likelihood (ML) and Bayesian approach (BA) methods were used to reconstruct the phylogenetic trees. The resolved ML and BA trees showed similar topology, thus revealing two major evolutionary lineages within the Sciaenidae family, namely, Western Atlantic (WA) and Eastern Atlantic–Indo–West Pacific (EIP). The WA group included four species belonging to four genera: *Cynoscion nebulosus*, *Equetus punctatus*, *Sciaenops ocellatus*, and *Micropogonias undulatus*. Meanwhile, the EIP group formed one monophyletic clade, harboring eight species (*Argyrosomus regius*, *A. japonicus*, *Pennahia anea*, *Nibea albiflora*, *Miichthys miiuy*, *Collichthys lucidus*, *Larimichthys polyactis*, and *L. crocea*) from six genera. Our results indicated that the Western Atlantic (WA) group was more ancient in the studied species, while the Eastern Atlantic–Indo–West Pacific (EIP) group was a younger group. Within the studied species, the genera *Collichthys* and *Larmichthys* were the youngest lineages, and we do not suggest that *Collichthys* and *Larmichthys* should be considered as one genus. However, the origin of the Sciaenidae family and problems concerning the basal genus were not resolved because of the lack of genomes. Therefore, further sampling and sequencing efforts are needed.

## 1. Introduction

The Sciaenidae family, commonly known as croakers or drums, is a unique group of fish that produce characteristic sounds through the sonic muscles and swim bladder. It is considered that the generated sounds may be used by those fish for communication and sex identification during reproduction [1,2]. In addition, the Sciaenidae family includes 68 genera that comprise approximately 293 extant species [3,4]. The members of this family are distributed across temperate and tropical seas worldwide, thus displaying a high level of biodiversity in the Atlantic, Indian, and Western Pacific regions. Moreover, this family harbors some commercially important species in China, such as the large yellow croaker, *Larimichthys crocea*, and the small yellow croaker, *Larimichthys polyactis* [5]. Their large economic and scientific values have always attracted great interest from biogeographers in studying the origins, patterns, and phylogeny of Sciaenidae fish [6]. Nevertheless, uncertainty still exists regarding the phylogenetic relationships of the family, primarily because of limitations in the available morphological and genetic data [7].

Thus far, Chao proposed an initial phylogenetic classification of the Sciaenidae family based on the anatomical features of the swim bladder [8]. In turn, Sasaki was the first to discuss the origin of the Sciaenidae family and proposed its monophyly based on morphology. The author also postulated the hypothesis that the origin of Sciaenidae occurred in the New World and diversified via an eastward and/or westward expansion path [9]. However, that hypothesis was merely based on morphological data, thus lacking strong evidence.

Phylogenetic studies based on molecular data have become a hot topic since the 21st century. At first, a limited set of mitochondrial and nuclear genes were used as markers in the phylogeny of Sciaenidae fish [10,11,12,13,14,15,16,17]. However, they may not be reliable because of the introgression, saturation, or selection of mitochondrial genes [18]. In addition, the phylogenetic analysis based on the different genetic markers often results in different evolutionary inferences. For example, Lo et al. performed a comprehensive study based on the cyt *b*, *COⅠ*, *RAG1*, *RH*, *EGR1*, and *EGR2B* genes of Sciaenidae and revealed that the genus *Argyrosomus* was the basal genus of the Western Pacific species [6]. However, the study based on mtDNA genomes supported that the genus *Argyrosomus* and the genus *Sciaenops* formed a sister lineage, whereas the genus *Miichthys* occupied the basal position within the West Pacific lineage [17]. Therefore, to date, the phylogenetic status of the Sciaenidae family has not been ambiguously resolved, mainly because of the application of the limited amount of genetic data for the study. Therefore, in this study, we summarized three main problems of the Sciaenidae phylogeny that still require solutions: (1) Which genus occupies the basal position within the family? (2) Which genus is the most ancestral within the high-diversified Indo–Western Pacific Sciaenidae group? (3) Should the *Collichthys* and *Larimichthys* genera be merged into one genus? Thus far, none of the listed questions have been answered by phylogenomic analysis.

Recently, phylogenomic analysis has become the primary trend to solve problems in phylogenetic studies [19]. The cost of genome sequencing has dramatically reduced along with the development of next-generation sequencing technologies, thereby enabling robust phylogenetic reconstructions based on genome-wide data [20]. Within the past two decades, almost a thousand fish genomes have been sequenced, thus providing vast usable data for large-scale phylogenetic studies [21]. Among plenty of available genomic data, exon genetic information is widely used because of the easy alignment and high level of conservation among species, which makes it particularly robust in phylogenetic studies [22]. For example, Malmstrøm et al. initially established the firm phylogenetic relationships of the teleost and gadiform using a data set of 567 homologous exons from 111 genes; thus, this genomic analysis revealed the evolution of immune genes in the teleost fish [23]. Hughes et al. successfully reconstructed the life-tree of 303 ray-finned fish by applying a total of 1105 single-copy orthologous data sets [7].

Moreover, we applied the available genomic and transcriptomic data to perform a phylogenomic analysis and reconstruct more reliable and comprehensive phylogenetic relationships within the Sciaenidae family. Our phylogenomic study provides a reference to resolve the controversy and origin of the Sciaenidae fish.

## 2. Materials and Methods

### 2.1. Data Collection and Transcriptome De Novo Assembly

The genomic data of 12 species from 10 genera of the Sciaenidae family and two outgroup species were collected from the NCBI GeneBank database (SRR5903997, SRR10001351, SRR5997754, SRR12344258, and SRR13555243) (Table 1). *Coreoperca whiteheadi* (from the GeneBank database) and *Sillago sinica* [24] were used as the outgroup. The Trimmomatic software (version 0.36) was used to clean the raw RNA-seq data, removing reads with sequencing adapters, unknown nucleotides (N ratio > 10%), and low-quality reads (quality scores < 20) [25]. Meanwhile, the software Trinity (version 2.8.5) [26] was used for the transcriptome de novo assembly of each selected species, and the parameters were as follows: -genome_guided_max_intron 10,000. Then, we used a Perl script (namely “extract_longest_isoforms_from_TrinityFasta.pl”, which can be found in Appendix A) with the default to extract the longest unigene from the resulting Trinity file. The longest unigene will be used for the next phylogenetic study.

### 2.2. Exon Captures and Phylogenetic Inference

Sixteen Sciaenidae genomes, five transcriptomes (longest extracted unigene files), and two outgroup genomes were included for the phylogenomic analysis. A data set of 1105 single-copy orthologs (conserved exon markers > 200 bp) by comparing eight well-annotated fish genomes was obtained from Hughes et al. [7]. The gathered data set was subjected to paralogy filtering through gene genealogy interrogation to confirm the orthologous state of each locus. Meanwhile, the collected genomes and unigene files were used to search against this 1105 single-copy orthologous data set using the nHMMER program within the HMMER (v.3.3.2) through the hidden Markov model (HMM) [27]. Subsequently, the identified exons were aligned and concatenated by Perl script (Appendix A). Gblocks v0.91b was used to eliminate unaligned loci [28]. Furthermore, the ProtTest software (v.3.4.2; Oxford University Press, New York, NY, USA) was used to obtain the best-fitting model for gene alignment, where the best model was BLOSUM62+I+G+F based on both AIC and BIC [29]. For the maximum likelihood (ML) analysis, a total of 1000 bootstrap replicates were performed for all of the concatenated nucleotide sequences GTRGAMMAI model implemented in the RAxML software (v.8.2.12; Oxford University Press, New York, NY, USA) [30]. Then, the Bayesian analysis (BA) was also carried out by Mr. Bayes 3.2.7 under the BLOSUM62+I+G+F model [31]. Hence, 100,000 metropolis-coupled Markov chain Monte Carlo (MCMCMC) generations were applied for this purpose, where every 100th generation was sampled and then the first 25% of the sampled generations were discarded as burn-in.

## 3. Results

### Phylogenetic Analysis of the Studied Species

Herein, a filtered, concatenated, and aligned data set of 69,098 bp, covering 309 shared single-copy orthologs, was used for the phylogenetic analyses. The reconstructed ML and BA trees were highly congruent and well supported by high bootstrap pseudo-replicate scores and significant posterior probabilities (>95%) for most nodes (Figure 1 and Figure 2).

In our study, the reconstructed phylogenetic trees indicated the presence of two main evolutionary lineages within the Sciaenidae family, namely: (1) West Atlantic (WA) and (2) Eastern Atlantic–Indo–West Pacific (EIP) cluster, and the former was more ancient on both trees. The WA cluster included four species belonging to four genera: *C. nebulosus*, *E. punctatus*, *S. ocellatus*, and *M. undulatus*. Within the WA group, the ML tree resolved that a lineage comprising *M. undulatus* and *S. ocellatus* has a closer relationship with the genus *Equetus*. In turn, the BA tree displayed a closer relationship between this lineage and the genus *Cynoscion.* Both trees indicated that EIP represents a monophyletic lineage that includes congruent phylogenetic topology within the EIP lineage, which was monophyletic and included eight species representing six genera: *A. regius*, *A. japonicus*, *P. anea*, *N. albiflora*, *M. miiuy*, *C. lucidus*, *L. polyactis*, and *L. crocea*. Both trees supported close evolutionary relationships between *M. undulatus* and *S. ocellatus*. Six *L. crocea* individuals were well separated as a single distinct lineage in the topology of both resolved trees.

## 4. Discussion

### 4.1. Representativeness of Studied Species

Herein, we collected all available genome-wide data on Sciaenidae to reconstruct reliable and comprehensive phylogenetic relationships within the Sciaenidae family. Although only 12 species that belong to 10 genera were included, our sampling strategy well represented the high diversity of Sciaenidae. First, our taxa sampling covered the main habitat of Sciaenidae as the studied species were distributed from tropical to subfrigid zones, including the Atlantic, Indian Ocean, and Western Pacific (Table 1). Widely distributed species such as *A. regius* and regional species such as *L. crocea* examined in our study were also included [32,33]. Finally, the studied species represent large behavioral and morphological diversities regardless of living in a homogeneous habitat. For example, two sibling species in the Western Pacific, *L. crocea* and *L. polyactis*, had large differences in their behavior such as spawning season and migratory route [33,34]. Although most representative species of the Sciaenidae family were included in our study, more genomes are still required for further investigation.

### 4.2. Phylogeny of Sciaenidae

For the first time in this study, we reconstructed the phylogenetic relationship within the Sciaenidae family based on genome-wide data. The results of our study indicate that the Sciaenidae family is represented by two main phylogenetic lineages, namely: (1) Western Atlantic (WA) and (2) Eastern Atlantic–Indo–West Pacific (EIP) clusters. The WA phylogenetic lineage resolved in our study was represented by four species belonging to four genera, namely: *C. nebulosus*, *E. punctatus*, *S. ocellatus*, and *M. undulates*. Herein, the sister relationship between the genera *Sciaenops* and *Micropogonias* was revealed, which is concordant with the previous results reported by Lo et al. based on the analysis of the multi-gene data set of several Sciaenidae fish species [6]. However, our study did not provide an answer as to which genus represents the basal genus within the WA cluster because of the non-correspondence in the resolved ML and BA topologies, which is most probably caused by the limited taxa sampling. In addition, the Western Atlantic Sciaenidae species were supported as the more ancient group, which conformed to the fossil evidence [35,36]. However, we did not resolve the problems concerning the origin of Sciaenidae due to the lack of samples. Thus, further studies should focus on the Western Atlantic species with the application of a larger number of genes for the comprehensive phylogeny reconstruction of this lineage.

The EIP phylogenetic lineage was represented by eight species belonging to six genera, namely: *Argyrosomus regius*, *A*. *japonicus*, *Pennahia anea*, *Nibea albiflora*, *Miichthys miiuy*, *Collichthys lucidus*, *Larimichthys polyactis*, and *L*. *crocea*. ML and Bayesian trees suggested the same phylogenetic relationships. Based on our results, the genus *Argyrosomus* was more ancient within the EIP clade. However, our results did not resolve the lack of samples for the problem of the basal genus of the EIP group. The study based on the multi-gene dataset indicated that the genus *Totoaba* was more ancestral to this clade, and the genus *Argyrosomus* branched after it [6]. Moreover, our study also indicated the close evolutionary relationships between *Miichthys*, *Collichthys*, and *Larimichthys* genera, wherein all included species from those genera formed one monophyletic lineage within the EIP cluster. However, the phylogenetic analysis we carried out did not support previous studies based on analysis of mitochondrial *COI*, cyt *b*, and 16S rRNA genes, which suggested that *Collichthys* and *Larmichthys* should be considered as one genus [37]. Nevertheless, other species from the genus *Collichthys* (like *C. niveatus*) were not included in our study because of the lack of genomes. Therefore, more sequencing efforts are needed in further phylogenetic analysis to confirm this conclusion and resolve the remaining problems.

## 5. Conclusions

In this study, genome-wide data were used in the phylogenetic analysis of the Sciaenidae family. The results indicated that the Sciaenidae family harbors two major evolutionary lineages: (1) the Western Atlantic (WA) and (2) the Eastern Atlantic and Indo–Western Pacific (EIP) lineages. Thus, the results of our phylogenetic study provide a reference to resolve the existing controversy of the Sciaenidae family: (1) the Western Atlantic species is the more ancient group of the family Sciaenidae; however, the origin and the specific basal genus were not confirmed. (2) The genus *Argyrosomus* is the more ancient genus of the studied Indo–Western Pacific Sciaenidae species; however, the problem concerning the basal genus was not resolved because of the lack of samples. (3) The genera *Collichthys* and *Larimichthys* should not be merged into one genus. Despite our study unraveling phylogenetic relationships among the fish belonging to the Sciaenidae family, some shortages remain in the overall evolutionary picture of the family because of the gaps in the available data. Therefore, further taxonomic sampling and sequencing efforts are needed to determine more comprehensive phylogenetic relationships within the Sciaenidae family.

## Figures and Tables

**Figure 1 animals-12-03386-f001:**
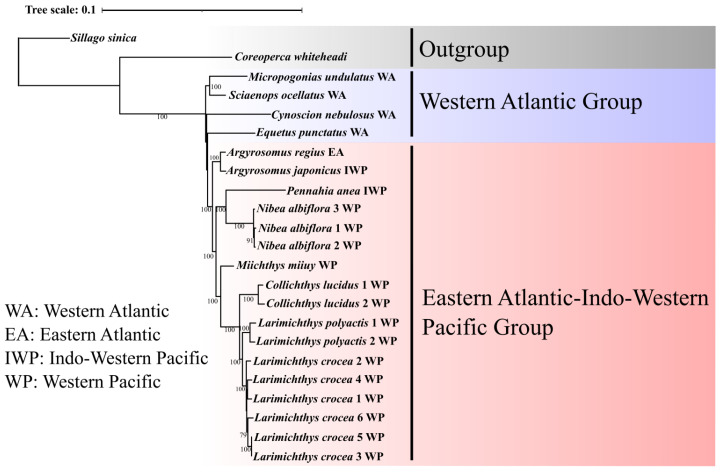
ML tree based on concatenated nucleotide data. Numbers at each node indicate the degree of support (only bootstrap values that are >70% are shown).

**Figure 2 animals-12-03386-f002:**
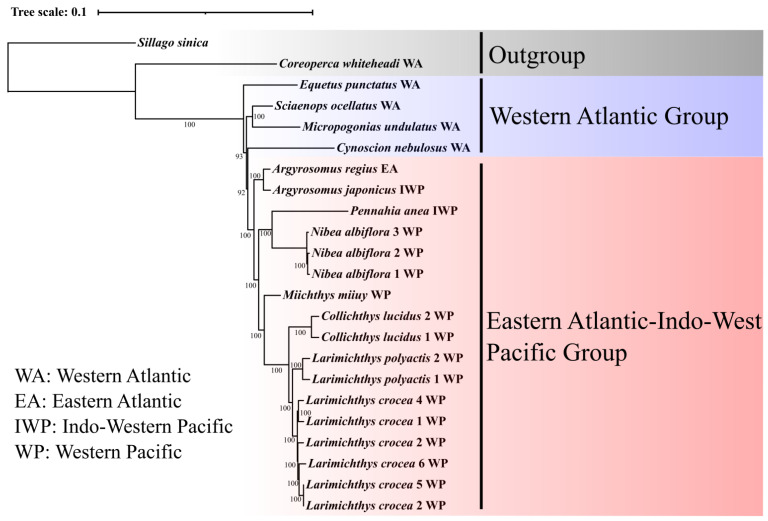
BA tree based on concatenated nucleotide data. Numbers at each node indicate the degree of support (only bootstrap values that are >70% are shown).

**Table 1 animals-12-03386-t001:** Genomic and transcriptome data information for the species examined in this study.

Scientific Name	Distribution *	Data Type	Total Sequence (Mb)	Contig N50 (bp)	Source
**1** ***Sciaenops***					
*Sciaenops ocellatus*	Western Atlantic	Genome	648,634	40,922	GCA_14183145.1
**2** ***Micropogonias***					
*Micropogonias undulatus*	Western Atlantic	Transcriptome	157,291	1304	SRR10001351
**3** ***Equetus***					
*Equetus punctatus*	Western Atlantic	Transcriptome	90,036	1976	SRR5997754
**4** ***Cynoscion***					
*Cynoscion nebulosus*	Western Atlantic	Transcriptome	46,823	953	SRR12344258
**5** ***Argyrosomus***					
*Argyrosomus regius*	Eastern Atlantic	Transcriptome	812,965	3012	SRR5903997
*Argyrosomus japonicus*	Indo–West Pacific	Genome	79,196	1984	GCA_15710095.1
**6** ***Pennahia***					
*Pennahia anea*	Indo–West Pacific	Transcriptome	812,965	2223	SRR13555243
**7** ***Nibea***					
*Nibea albiflora* 1	Northwest Pacific	Genome	619,301	1212	GCA_014281875.1
*Nibea albiflora* 2	Northwest Pacific	Genome	574,466	34,769	GCA_900327885.1
*Nibea albiflora* 3	Northwest Pacific	Genome	595,677	3712	GCA_902410095.1
**8** ***Miichthys***					
*Miichthys miiuy*	Northwest Pacific	Genome	627,579	20,386	GCA_001593715.1
**9** ***Collichthys***					
*Collichthys lucidus* 1	Western Atlantic	Genome	877,615	2911	GCA_004119905.2
*Collichthys lucidus* 2	Western Atlantic	Genome	671,995	137,747	GCA_009852395.1
**10** ***Larimichthys***					
*Larimichthys crocea 1*	Northwest Pacific	Genome	65,794	16,979	GCA_000972845.1
*Larimichthys crocea 2*	Northwest Pacific	Genome	648,391	51,577	GCA_000742935.1
*Larimichthys crocea 3*	Northwest Pacific	Genome	744,270	4085	GCA_003711585.2
*Larimichthys crocea 4*	Northwest Pacific	Genome	721,257	1576	GCA_003845795.1
*Larimichthys crocea 5*	Northwest Pacific	Genome	7,441,011	3905	GCA_004352675.2
*Larimichthys crocea 6*	Northwest Pacific	Genome	689,173	9930	GCA_90024615.1
*Larimichthys polyactis* 1	Northwest Pacific	Genome	649,447	172,294	GCA_10119295.1
*Larimichthys polyactis 2*	Northwest Pacific	Genome	706,148,513	1,208,045	GCA_18985215.1
**Outgroup**					
*Sillago sinica*		Genome	536,576	2662	Xu et al. (2018) [25]
*Coreoperca whiteheadi*		Genome	712,457	76,587	GCA_011952105.1

* Distribution data derived from FishBase: https://www.fishbase.org (accessed on 30 September 2022).

## Data Availability

Not applicable.

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
