# Peer review of "The Phylogenetic Relationships of the Family Sciaenidae Based on Genome-Wide Data Analysis"

_animals, 2022, doi:10.3390/ani12233386_

Round 1

Reviewer 1 Report

Dear Editors,

Dear Authors,

The reviewed study entitled: “The Phylogenetic Relationships of The Family Sciaenidae Based on Genome-wide Data Analysis” represents quite interesting approach to phylogenetic research of fish belonging to Sciaenidae family. The applied methods are suitable and correct. However, manuscript requires considerable improvements. Manuscript is frequently written imprecisely, and a lot of terms and sentences require clarification. Language presentation is also not perfect and requires deep improvement. I recommend professional language correction. The introduction chapter does not sufficiently introduce reader to the controversies, questions, and problems in the phylogeny of Sciaenidae fishes. Materials and Methods requires better description because it is not reproducible for anyone but authors. Results sometimes require more clear description. Discussion chapter is composed by a lot of information that replicate information from the introduction. Conclusion chapter is not adequate to the content of the manuscript.

In conclusion, I do not recommend the reviewed manuscript in the present form for possible publication in the Animals periodical. Authors should make corrections and try again after major revision. All remarks, questions and fixes were placed in the attached pdf file (yellow highlights contain fixes and sentence suggestions, while red highlights contain comments and questions).

Thank you for another interesting manuscript that I could review!

Author Response

Dear Reviewer:

Thank you for your hard work and careful comments. On behalf of my co-authors, we are very grateful to you and thank you for giving us an opportunity to revise our manuscript.

We have studied your comments carefully and tried our best to revise our manuscript according to the comments.

Firstly, we revised the manuscript according to your yellow highlights. We thank you for your serious comments. Revised manuscript have been submitted, please check it.

Secondly, we revised the manuscript according to your red highlights. The following are the responses and revisions, I have made in response to the reviewer’s questions and suggestions on an item-by-item basis.

Abstract

In this section, we have read your comments carefully and revised the unclear sentences to more about discussion, the details are as follows.

Review PDF file: line 17-19

Reply: Thank you for your comments. We have modified the materials and methods section to improve our manuscript. The “analysis supported by genome-wide data”, that's what we meant. So, we have revised “Genome-wide data supported” to “Phylogenomic”. We summarized more of the discussion, unclear sentences have been avoided. (Revised PDF file: Line 26-32 )  

Review PDF file: line 19-20

Reply: Thank you for your advice. The remained problems were summarized in introduction section to improve this place (Revised PDF file: Line 71-75 ). And we gave the answer of those problems in discussion section. We delete unclear sentence. Please check.

Review PDF file: line 20-21

Reply: Thank you for your comments. This 1105 single-copy data set orthologous have not been used in resolving the phylogeny of the family, and we obtained well-supported phylogenetic trees. So, we said “ Our results well verified...”. We have avoided this unclear sentence.

Introduction

Review PDF file: line 44-45 and 51-52

Reply: Thank you for your advice. Reference number have benn revised. We've simplified the expression here, only hypothesis 2 was retained. Please check (Revised PDF file: Line 54-58 ).

Review PDF file: line 55-56 and 60-61

Reply: Thank you for your comments. We made major changes here, Some disadvantages of mitochondrial genes and examples are mainly mentioned to support our views. Please check (Revised PDF file: Line 60-71).

Review PDF file: line 69-71

Reply: Thank you for your advice. We have revised the expression. (Revised PDF file: Line 84-87 )

Review PDF file: line 73

Reply: Thank you for your advice. We have added some details of the paper as suggested by you, please check (Revised PDF file: Line 87-90).

Review PDF file: line 73-77

Reply: Thank you for your comments. We have simplified the expression here as you suggested  (Revised PDF file: Line 90-91 ). Additionally, we added a sentence to summary this paragraph (Revised PDF file: Line 91-92 ). Please check.

Review PDF file: line 79-80

Reply: Thank you for your comments. We have deleted this sentence.

Review PDF file: line 85-88

Reply: Thank you for your advice. We have deleted this sentence and revised manuscript. Please check (Revised PDF file: Line 93-99 ).

Materials and methods

At the suggestion of the editor, we made “Materials and methods” as the second section. We revised and improved this section, making it easier for other researchers to replicate our experiment. We have submitted our detailed script in supplements, namely detailed_ptorocol.txt. The details are as follows.

Review PDF file: line 215

Reply: Thank you for your comments. We used a perl script with default to extract the longest unigene, this perl script have been provided in supplement. Please check (Revised PDF file: Line 109-111 )

Review PDF file: line 216

Reply: Thank you for your advice. In this step, the extracted longest unigene just be aligned and be used to next phylogenomic analysis, we have revised this place and detailed steps will be provided in supplement. Please check (Revised PDF file: Line 109-112 )

Review PDF file: line 219-221

Reply: Thank you for your comments. In this study, we used this data set of 1105 single-copy orthologous, which have been well tested. This data set was not provide by us, we obtained this data set from a published paper (Hughes, L. C.; Ortí, G.; Huang, Y.; Sun, Y.; Baldwin, C. C.; Thompson, A. W.; et al., Comprehensive Phylogeny of Ray-finned Fishes (Actinopterygii) Based on Transcriptomes and Genomic Data. Proceedings of the National Academy of Sciences of the United States of America 2018, 115, 6249-6254.) In that study, authors provided this well tested  data set and methods to let others use for performing phylogenomic analysis, so, we used the scripts provided by Hughes and some custom scripts to search the shared exons of our studied species form this data set and used for phylogenomic analysis. We have revised this place to make readers easier to understand. (Revised PDF file: Line 115-119)

Review PDF file: line 221-222

Reply: Thank you for your advice. The nHMMER program was used to find the single-copy orthologous from studied species, we used the python script to seach, which can be found in the supplement from Hughes et al. (2018), namely hmmer2fasta-single.py. (Revised PDF file: Line 119-121)

Review PDF file: line 224-225

Reply: Thank you for your comments. The detailed protocool can be found in Hughes et al. (2018), and we can provide our custom script, there are several ways to used the methods provied by Hughes et al. (2018). We just provide our custom method.

Review PDF file: line 225-226

Reply: Thank you for your advice. Gblocks was used to remove redundant, we have revised the manuscript, please check (Revised PDF file: Line 122-123).

Review PDF file: line 226

Reply: We have submitted our detailed script in supplements, namely detailed_ptorocol.txt

Review PDF file: line 228-229

Reply: Thank you for your comments. We revised the manuscript and the exact script can be found in detailed_ptorocol.txt.

Results

Review PDF file: line 97

Reply: Thank you for your advice. Reference have been added. (Revised PDF file: Line 140)

Review PDF file: line 98

Reply: Thank you for your comments. We used a perl script from Trinity software to obtain the number of contig N50 of our assembly results. (Revised PDF file: Line 142-143)

Review PDF file: line 110

Reply: Thank you for your advice. We have revised this place, please check (Revised PDF file: Line 154-155).

Review PDF file: line 113

Reply: Thank you for your comments. The details have been added here, please check (Revised PDF file: Line 157-158 ).

Discussion

Figure 3 have been deleted for readers could find in references.

Review PDF file: line 147-148

Reply: Thank you for your advice. Firstly, eight species from the Eastern Atlantic, Indian Ocean and Western Pacific formed one clade, so we made them as one group. Secondly, four Western Atlantic species were made as one group although they didn’t formed one clade, but in terms of geographical distribution, this may be both concise and reasonable. Finally, we also consider multiple groupings, it is certainly doesn't affect the diversity of the Sciaenidae family, however, the genus Argyrosomus can be divided as one clade. So, combing various reasons, we selected two-groups to explain our results. We revised the manuscript, please check (Revised PDF file: Line 195 ).

Review PDF file: line 165-169

Reply: Thank you for your comments. We revised the manuscript and put two problems here, please check (Revised PDF file: Line 215-224 ).

Review PDF file: line 170-173

Reply: Thank you for your advice. We added the reason about the phylogeney of the genus Pennahia and added reference, please check (Revised PDF file: Line 225-229 ).

Review PDF file: line 175-176

Reply: Thank you for your comments. We referenced a study here to explain the merging of two genera, please check (Revised PDF file: Line 231-234 ).

Review PDF file: line 176-177

Reply: Thank you for your advice. We added the reason that this two genera lacking for genomes, so we said “further studies...”, please check (Revised PDF file: Line 234-236 ).

Review PDF file: line 184

Reply: Thank you for your comments. We reviesed “but lacked test” to “ the fossil data was not considered in his analysis”, please check (Revised PDF file: Line 240-241 ).

Review PDF file: line 186

Reply: Thank you for your advice. We revised here, the American fossil appeared earlier than Asian fossil, please check (Revised PDF file: Line 245-247 ).

Review PDF file: line 189

Reply: Thank you for your comments. We revised the manuscript to make readers easily to understand. Please check (Revised PDF file: Line 243-249 ).

Review PDF file: line 190

Reply: Thank you for your advice. Lo et al. (2015) studied 52 genera (about 74%) and 93 species, so we said “comprehensive”, we have revised the manuscript here, please check (Revised PDF file: Line 250 ).

Review PDF file: line 196-201

Reply: Thank you for your comments. This section have been deleted.

Conclusion

Review PDF file: line 235-237

Reply: Thank you for your advice. We have deleted some sentences and removed references. (Revised PDF file: Line 257 ).

Review PDF file: line 238 & 245

Reply: Thank you for your comments. We added the answer here (Revised PDF file: Line 266-271 ) for the summarized problems in introduction. Please check.

Supplementary Materials

New supplement contains detailed scripts have been updated.

Thanks again!

Reviewer 2 Report

The scope of manuscript is interesting enough as the study used comprehensive genome-wide markers to investigate the phylogenomics of a group of fish with high economic significance. However, at the current stage, this manuscript seems more like a part of phylogenomic section of the paper presenting a new genome, which is recently published by the same team (Zhao et al., 2021). As a phylogenomic paper on its own, the manuscript is still relatively shallow in terms of two aspects.

One aspect is the comprehensiveness of sampling. As the study would like to tentatively address questions like “the origin of Sciaenidae”, samples from only 10 out of 68 genera is insufficient. Therefore, the results are also turns out “unsurprising” with limited new information added to the current understanding of the taxonomy in the Family Sciaenidae. Although there might be limited availability of transcriptomes and genomes, authors may consider blasting “legacy markers” in the GenBank from the exon-capture data of more genera.

The other aspect is the depth of analyses. In the manuscript, all the genome-wide markers are concatenated without partitioning to reconstruct the phylogenies, which assumes all codon positions of all the markers sharing the same evolutionary history. To fully utilize the versatility of genome-wide exons in the exploration of phylogeny, I would suggest the authors to refer to the workflow of recently published papers for the paralogues check in legacy markers, codon partitioning in concatenation-based analyses (Arcila et al., 2021) and species tree analyses (Astudillo-Clavijo et al., 2022).

References

Astudillo-Clavijo, V., Stiassny, M. L., Ilves, K. L., Musilova, Z., Salzburger, W., & López-Fernández, H. (2022). Exon-based phylogenomics and the relationships of African cichlid fishes: tackling the challenges of reconstructing phylogenies with repeated rapid radiations. Systematic Biology.

Arcila, D., Hughes, L. C., Meléndez-Vazquez, B., Baldwin, C. C., White, W. T., Carpenter, K. E., ... & Betancur-R, R. (2021). Testing the utility of alternative metrics of branch support to address the ancient evolutionary radiation of tunas, stromateoids, and allies (Teleostei: Pelagiaria). Systematic Biology, 70(6), 1123-1144.

Zhao, L., Xu, S., Han, Z., Liu, Q., Ke, W., Liu, A., & Gao, T. (2021). Chromosome-Level Genome Assembly and Annotation of a Sciaenid Fish, Argyrosomus japonicus. Genome biology and evolution, 13(2), evaa246.

Author Response

Dear Reviewer:

Thank you for your hard work. On behalf of my co-authors, we are very grateful to you and thank you for your advice.

We have studied the references provided from you carefully and discussed our shortcomings. Firstly, about the paper of Zhao et al. (2020), the first chromosome-level genome of Argyrosomus japonicus was published in this study. And they also used single-copy orthologous to construct the phylogenetic relationships, but they used Argyrosomus japonicus, two Sciaenidae species and some model or well-known species’ genomes. Coincidentally, Sillago sinica confirmed was also selected in our study, but we used it as outgroup. Additionally, our study mainly studied the phylogeny of the family Sciaenidae, Zhao et al. mainly studied the genome information. Although Zhao et al. and we both used single-copy orthologous as markers to perform phylogenetic analysis, but we used a well-tested data set, which passed the paralogy filtering through the gene genealogy interrogation (Hughes et al. 2018).

Secondly, we studied the phylogenetic methods studies provided from you, it's really worth learning from, but due to lack of time, the methods may be used in further studies.

Thanks again!

Reference

Zhao, L.; Xu, S.; Han, Z.; et al., Chromosome-Level Genome Assembly and Annotation of a Sciaenid Fish, Argyrosomus japonicus. Genome Biology and Evolution 2021, 13, (3), evaa246.

Hughes, L. C.; Ortí, G.; Huang, Y.; Sun, Y.; Baldwin, C. C.; Thompson, A. W.; et al., Comprehensive Phylogeny of Ray-finned Fishes (Actinopterygii) Based on Transcriptomes and Genomic Data. Proceedings of the National Academy of Sciences of the United States of America 2018, 115, 6249-6254.

Reviewer 3 Report

In this study, the authors provide the phylogenetic relationships of the family Sciaenidae based on genome-wide data on12 species. I found the manuscript interesting and the findings important in answering knowledge gaps in the phylogeny of Sciaenidae. However, the big problem is why authors analyze the phylogenetic relationships of  family Sciaenidae based on only 12 species in this study, becasue in the family more than 12 species genome data have been reported 

Please also have the manuscript proofread before resubmitting as there are many sentence structure and writing errors within the manuscript. In my opinion, this work would be valuable after important changes based on the following recommendations:

line 8-10, the sentence is not very clear, pls rewrite it; 18 species were mentioned here, but in the text below only 12 species, pls check and modify;

line 30-31, the reference is too old, I think there is not only 292 species, should be updated;

line 32, delete temperate;

line 36-38, a reference should be added into the sentence;

line 47,49, the symbols before (2) and (3) should be same;

line 92-93, "NCBI database" should be modified as "GenBank database";

table 1, in the Source of Sillago sinica, a cited reference should be put in there, a website link is not well;

Figure 1 and 2, pls display the outgroup.

figure 3, no need to exhibit the reported figures in the present study, readers can find the original figures from the references;

Author Response

Dear Reviewer:

Thank you for your comments. On behalf of my co-authors, we are very grateful to you for giving us an opportunity to revise our manuscript.

We have studied your comments carefully and tried our best to revise our manuscript according to the comments. The following are the responses and revisions, I have made in response to the reviewer’s questions and suggestions on an item-by-item basis.

line 8-10, the sentence is not very clear, pls rewrite it; 18 species were mentioned here, but in the text below only 12 species, pls check and modify;

Reply:Thank you for your comments. We checked this place carefully, the reason for the ambiguity is that we collected 18 genomes of 7 Sciaenidae spceies and 2 outgroup species, some species including multiple genomes from different source, like Larimichthys crocea, which have 6 genomes. With 5 transcriptomes of 5 Sciaenidae spceies, there were 12 Sciaenidae spceies and 2 outgroup species were covered in our study. We revised the manuscript to avoid the ambiguity. (Revised Manuscript PDF file, Line: 17-20)

line 30-31, the reference is too old, I think there is not only 292 species, should be updated;

Reply:Thank you for your comments. We recollected the data and finally determined that there were 293 species of the family. We have revised the manuscript and the list of references (Revised Manuscript PDF file, Line: 43).

Reference: Nelson, J. S., Fishes of the World, 4th ed. John Wiley and Sons 2006, New York.  

Froese, R.; Pauly, D., FishBase. World Wide Web electronic publication. 2022. Aviavilable from: http://www.fishbase.org.

line 32, delete temperate;

Reply:Thank you for your comments. We deleted “warm temperate” and changed it to "tropical". (Revised Manuscript PDF file, Line: 44 )

line 36-38, a reference should be added into the sentence;

Reply:Thank you for your comments. We add a reference here. In this study, The author reviewed the research status of the phylogeny of Sciaenidae, It basically covers major research from morphology to molecules, Therefore, we think this study can be referenced here. (Revised Manuscript PDF file, Line: 49 )

line 47,49, the symbols before (2) and (3) should be same;

Reply:Thank you for your advice. We have corrected the manuscript, hypothesis (1) and (3) were deleted to make the expression more concise. (Revised Manuscript PDF file, Line: 53-58 )

line 92-93, "NCBI database" should be modified as "GenBank database";

Reply:Thank you for your advice. We have revised the manuscript. (Revised Manuscript PDF file, Line: 102 and 134-135 )

table 1, in the Source of Sillago sinica, a cited reference should be put in there, a website link is not well;

Reply:Thank you for your comments. We have revised this place, the reference was put in here. (Revised Table 1 )

Figure 1 and 2, pls display the outgroup.

Reply:Thank you for your advice. We have revised figures and new figures have been used in the revised manuscript. (Revised figure 1 and 2)

figure 3, no need to exhibit the reported figures in the present study, readers can find the original figures from the references;

Reply:Thank you for your advice. We have deleted figure 3 and modified expression to make more concise.

Thanks again!

Round 2

Reviewer 1 Report

Dear Editor,

Dear Authors,

The reviewed manuscript entitled: “The Phylogenetic Relationships of The Family Sciaenidae Based on Genome-wide Data Analysis” has been significantly improved according to the Reviewers recommendations. Taking into account the substantive content, the manuscript is correct and the applied methods adequate to the specified aims. The reviewed manuscript requires only improvements of language presentation. The most evident corrections I put by myself. I also recommend professional language corrections before publishing. All fixes and remarks are included in the attached file (yellow highlights contain fixes and sentence suggestions, while red highlights contain comments and questions).

Thank you for another interesting manuscript that I could review!

Author Response

Dear reviewer,

Thank you for your revision suggestion again. We read the review carefully and revised the manuscript according to it. We revised the manuscript according to yellow highlights, and our reply to red highlights is as follows.

Review PDF file: line 114

Reply: Thank you for your comments. Yes, Hughes et al. performed this comparing. We revised this place, please check Revised PDF file: Line 114-116

Review PDF file: line 130

Reply: Thank you for your advice. We deleted the 3.1 section and move the Table 1 to Materials and Methods section (Revised PDF file: Line 99-100). The content about outgroup was also be moved to Materials and Methods section.

Review PDF file: line 137 and 143

Reply: Thank you for your comments. Table 2 have been deleted.

Review PDF file: line 144

Reply: Thank you for your advice. The 3.3 section have been revised to 3.1 and as the first section of Results, please check Revised PDF file: Line 133

Review PDF file: line 183

Reply: Thank you for your comments. Yes, we mean behavioral diversity, we have revised this place, please check Revised PDF file: Line 170

Thanks again!

Reviewer 2 Report

The manuscript is not really improved from the previous version.

The major concern is that the comprehensiveness of dataset is not sufficient for the many claims in the abstract and summary. For example, authors claim "Western Atlantic species were the most ancestral group of Sciaenidae". However, authors only present Western Atlantic species of genus Micropogonias and Cynoscion while omitting Eastern Pacific species. As for the other major findings that 1) "Argyrosomus to be basal genus of Eastern Atlantic, Indian Ocean and West Pacific Sciaenidae species" and 2) "sister lineage which contains the genera Pennahia and Nibea branched after the genus Argyrosomus" are also not sound and subject to the partial sampling (1). genus Umbrina in Eastern Atlantic can be more basal in Sciaenidae and 2). it is hard to say that Pennahia + Nibea is more basal than Miichthys + Collichthys + Larimichthys as many genera in both clades are not sampled). Therefore, the conclusion can be misleading.

In the conclusion, authors have toned down all the claims, which seems to me more relevant to the findings of the manuscript. I'd suggest revising the abstract and summary at least to the same style of conclusion. Moreover, the sampling effort is skew to the Indo-West-Pacific, therefore, claims on the phylogenetic positions should be more focused while avoiding speculating phylogeny outside the region. 

In general, authors need to adjust the manuscript significantly to make it a more regional-focused phylogenetic study that addresses previous phylogenetic questions in the Indo-Western Pacific group of Sciaenidae. The use of samples at Western Atlantic is more of a kind of outgroup of the focal clade (Indo-Western Pacific group) rather than evidence to claim origin.  

Author Response

Dear reviewer,

Thank you for your suggestion again. We read the review carefully and reply for your major concerns.

Firstly, we have collected all available genomes data. Our studied species could represent the major distribution of the family Sciaenidae. Selected species contains the most studied Sciaenidae species. Additionally, we deleted the discussion about the origin of Sciaenidae. We revised the discussion of the Eastern Atlantic species, avoiding the conclusion based on few samples. We made some major revision of the manuscript, please check. Finally, we revised the Abstract and Conclusion section, and made them as the same style. For details please check new revised manuscript.

Thanks again!